

# Isotopic characterization of nitrogen oxides (NOx), nitrous acid (HONO), and nitrate (NO3-(p)) from laboratory biomass burning during FIREX

**Jiajue Chai[1], David J. Miller[1,a], Eric Scheuer[2], Jack Dibb[2], Vanessa Selimovic[3], Robert Yokelson[3], Kyle J. Zarzana[4,5,b], Steven S. Brown[4,6], Abigail R. Koss[4,5,6,c], Carsten Warneke[5,6], Meredith Hastings[1]**

1. Department of Earth, Environmental and Planetary Sciences, and Institute at Brown for Environment and Society, Brown University, Providence, RI, USA
2. Institute for the Study of Earth, Ocean and Space, University of New Hampshire, Durham, NH, USA
3. Department of Chemistry, University of Montana, Missoula, USA
4. Chemical Sciences Division, NOAA Earth System Research Laboratory, Boulder, CO, USA
5. Cooperative Institute for Research in Environmental Sciences, University of Colorado, Boulder, CO, USA
6. Department of Chemistry, University of Colorado, Boulder, CO, USA
a. Now at: Environmental Defense Fund, Boston, MA, USA
b. Now at: Department of Chemistry, University of Colorado, Boulder, CO, USA
c. Department of Civil and Environmental Engineering, Massachusetts Institute of Technology, Cambridge, MA, USA

**Correspondence:** Jiajue Chai (jiajue_chai@brown.edu)





**Abstract**
New techniques have recently been developed to capture reactive nitrogen species for
accurate measurement of their isotopic composition. Reactive nitrogen species play
important roles in atmospheric oxidation capacity (hydroxyl radical and ozone formation)
and may have impacts on air quality and climate. Tracking reactive nitrogen species and
their chemistry in the atmosphere based upon concentration alone is challenging. Isotopic
analysis provides a potential tool for tracking the sources and chemistry of species such
as nitrogen oxides ($NO_x$ = NO + $NO_2$), nitrous acid (HONO), nitric acid ($HNO_3$) and
particulate nitrate ($NO_3^-$(p)). Here we study direct biomass burning (BB) emissions
during the  Fire Influence on Regional to Global Environments Experiment (FIREX, later
evolved into FIREX-AQ) laboratory experiments at the Missoula Fire Laboratory in the
fall of 2016.
An annular denuder system (ADS) developed to efficiently collect HONO for isotopic
composition analysis was deployed to the Fire Lab study. Concentrations of HONO
recovered from the ADS collection agree well with mean concentrations averaged over
each fire measured by 4 other high time resolution techniques, including mist
chamber/ion chromatography (MC/IC), open-path Fourier transform infrared
spectroscopy (OP-FTIR), cavity enhanced spectroscopy (CES), proton-transfer-reaction
time-of-flight mass spectrometer (PTR-ToF). The concentration validation ensures
complete collection of BB emitted HONO, of which the isotopic composition is
preserved during the collection process. In addition, the isotopic composition of $NO_x$ and
$NO_3^-$(p) from direct BB emissions were also characterized.
In 20 "stack" fires (direct emission within ~5 seconds of production by the fire) that
burned various biomass materials, $\delta^{15}N$-$NO_x$ ranges from -4.3 ‰ to +7.0 ‰, falling near
the middle of the range reported in previous work. The first measurements of $\delta^{15}N$-
HONO and $\delta^{18}O$-HONO in biomass burning smoke reveal a range of -5.3 – +5.8 ‰ and
+5.2 – +15.2 ‰ respectively. Both HONO and $NO_x$ are sourced from N in the biomass
fuel and $\delta^{15}N$-HONO and $\delta^{15}N$-$NO_x$ are strongly correlated ($R^2$ = 0.89, p<0.001),
suggesting $NO_x$ and HONO are connected via formation pathways.
Our $\delta^{15}N$ of $NO_x$, HONO and $NO_3^-$(p) ranges can serve as important biomass burning
source signatures, useful for constraining direct emissions of these species in
environmental applications. The $\delta^{18}O$ of HONO and $NO_3^-$ obtained here verify our
method is capable of determining oxygen isotopic composition in BB plumes. The $\delta^{18}O$
for both species in this study reflect the laboratory conditions (i.e. a lack of
photochemistry), and would be expected to track with the influence of ozone ($O_3$),
photochemistry and nighttime chemistry in real environments. The methods used in this
study will be further applied in future field studies to quantitatively track reactive
nitrogen cycling in fresh and aged Western US wildfire plumes.





## 1 Introduction

Biomass burning (BB), which occurs in both anthropogenic processes (e.g. cooking,
heating, prescribed) and natural wildfire (lightning ignited vegetation burning), is a
significant source of atmospheric reactive nitrogen species, including nitrogen oxides
($NO_x$ = NO + $NO_2$), nitrous acid (HONO), nitric acid ($HNO_3$), particulate nitrate ($NO_3^-$
(p)), organic nitrates, peroxyacyl nitrate (PAN) and ammonia ($NH_3$) that have major
impacts on air quality and climate from regional to global scales (Crutzen and Andreae,
1990). Globally, biomass burning emits ~6 Tg of nitrogen oxides ($NO_x$ = NO + $NO_2$) per
year, contributing at least 14% to total $NO_x$ emissions (Jaeglé et al., 2005), with large
interannual and seasonal variation due to fire frequency and intensity (Jaffe and Briggs,
2012). Primarily emitted $NO_x$ plays an important role in the photo-oxidation of volatile
and semi volatile organic compounds, which are present in high concentrations in BB
plumes, and strongly influences production of tropospheric ozone ($O_3$) and secondary
aerosols (Alvarado et al., 2015). In BB plumes, $NO_x$ can be converted to PAN, which can
be transported long distances (100s to 1000s of km) in lofted plumes before rereleasing
$NO_x$. Therefore, BB emitted $NO_x$ could widely influence air quality downwind for days
to weeks (Val Martín et al., 2006; Ye et al., 2016). In addition, $NO_x$ is also the major
photochemical precursor of $HNO_3$ and $NO_3^-$(p), which can be transported downwind and
mixed with anthropogenic emissions impacting air quality and ecosystem health
(Hastings et al., 2013).
HONO has been observed in BB plumes in both laboratory and field experiments, with
HONO mixing ratios in the range of ~5-33% of observed $NO_x$ (Akagi et al., 2012, 2013;
Burling et al., 2010, 2011; Keene et al., 2006; Liu et al., 2016; Roberts et al., 2010;
Selimovic et al., 2018; Yokelson et al., 2007, 2009). Photolysis of HONO is a major OH
precursor in the daytime; therefore HONO plays an important role in photochemical
aging of BB plumes and atmospheric oxidation capacity at regional scales (Alvarado and
Prinn, 2009; Liu et al., 2016; Tkacik et al., 2017; Trentmann et al., 2005). HONO has
been proposed as a significant OH source in BB plumes and the inclusion of HONO in
photochemical models could explain much of the uncertainty in the modeled $O_3$
(Alvarado et al., 2009; Alvarado and Prinn, 2009; Cook et al., 2007; Travis et al., 2016;
Trentmann et al., 2005).
Direct BB emission factor measurements of HONO and $NO_x$ exhibit significant
uncertainties due to limited observations and large spatial and temporal variability of
burning conditions, making it challenging to build an accurate inventory of BB emissions
relative to other major sources (Lapina et al., 2008). Emission factors vary and mainly
depend on 1) fuel nitrogen content (0.2 – 4% by mass), which is a function of vegetation
type, and 2) modified combustion efficiency (MCE = $\Delta[CO_2]/(\Delta[DCO] + \Delta[CO_2])$) that is
determined by combustion conditions including fuel moisture, fuel load, temperature,
relative humidity, wind speed, and other meteorological parameters (Burling et al., 2010;
Jaffe and Briggs, 2012; Yokelson et al., 1996). Additionally, the temporal evolution of
HONO in BB plumes varies greatly in different fires and relative contributions from
direct emission versus $NO_2$ conversion to HONO remains unclear. For instance,
significant concentrations of HONO and correlation between HONO and $NO_2$ have been
observed in aged plumes, indicating the importance of heterogeneous conversion of $NO_2$–
to–HONO on BB aerosols (Nie et al., 2015). By contrast, no evidence was found for



secondary HONO formation in a BB plume during the Southeast Nexus Experiment
(Neuman et al., 2016). HONO directly emitted from BB is important in constraining
HONO formed during plume aging and the total HONO budget (reducing uncertainties)
and increasing our understanding of HONO impacts on $O_3$ and secondary aerosol
formation downwind of BB regions.

In an effort to better understand reactive nitrogen emissions and chemistry, especially for
HONO, new techniques have been developed to analyze the isotopic composition of
various species. Stable isotopes provide a unique approach of characterizing and tracking
various sources and chemistry for a species of interest (Hastings et al., 2013). Fibiger et
al. (2014) developed a method to quantitatively collect $NO_x$ in solution as $NO_3^-$ for
isotopic analysis, which has been verified to avoid any isotopic fractionation during
collection in both lab and field studies. This allows for high-resolution measurement of
$\delta^{15}N$-$NO_x$ in minutes to hours depending on ambient $NO_x$ concentrations ($\delta^{15}N$ =
$[(^{15}N/^{14}N)_{sample}/(^{15}N/^{14}N)_{air-N2} - 1] \times 1000‰$, and $\delta^{18}O = [(^{18}O/^{16}O)_{sample}/(^{18}O/^{16}O)_{VSMOW} -$
$1] \times 1000‰$ where VSMOW is Vienna Standard Mean Ocean Water). $\delta^{15}N$ has also been
used to track gaseous $NO_x$ from a variety of major sources including emissions from
biomass burning (Fibiger and Hastings, 2016), vehicles (Miller et al., 2017), and
agricultural soils (Miller et al., 2018). Using this method, Fibiger et al. (2016)
systematically investigated BB $\delta^{15}N$-$NO_x$ from different types of biomass from around
the world in a controlled environment during the fourth Fire Lab at Missoula Experiment
(FLAME-4). $NO_x$ emissions collected both immediately from the BB source and 1-2
hours after the burn in a closed environment ranged from -7 to +12‰, and primarily
depended on the $\delta^{15}N$ of the biomass itself. BB emitted HONO isotopic composition has
never been measured before. Our recently developed method for HONO isotopic
composition analysis (Chai and Hastings, 2018) enables us to not only characterize $\delta^{15}N$
and $\delta^{18}O$ of HONO, but also explore the connection between $\delta^{15}N$-$NO_x$ and $\delta^{15}N$-HONO.

The Fire Influence on Regional to Global Environments Experiment and Air Quality
(FIREX-AQ) investigates the influence of fires in the western U.S. on climate and air
quality, via an intensive, multi-platform, campaign. The first phase of FIREX-AQ took
place at the US Forest Service Fire Sciences Laboratory (FSL) in Missoula, Montana, in
the fall of 2016, where we measured $\delta^{15}N$-$NO_x$, $\delta^{15}N$-HONO, $\delta^{18}O$-HONO, $\delta^{15}N$-$NO_3^-$
(p), $\delta^{18}O$-$NO_3^-$(p) and $\delta^{15}N$-biomass in 20 "stack burns" of a variety of fuels
representative of northwestern North America. Here we report on the results and explore
relationships between the isotopic composition of these nitrogen oxides, as well as the
corresponding mixing ratios for HONO that were concurrently measured by a variety of
techniques. This work offers characterization and quantification of BB source signatures
of these species, which can be applied in the interpretation of observations in future field
studies.



## 2. Experimental details

### 2.1 FIREX Fire Science Laboratory design

The room for controlled BB experiments is 12.5 × 12.5 m × 22 m, with a continuously weighed fuel bed at the center of the room. The combustion exhaust was vented at a constant flow rate (~3.3 m s$^{-1}$) through a 3.6 m diameter inverted funnel followed by a 1.6 m diameter stack, and collected at a platform 17 m above the fuel bed via sampling ports that surround the stack, resulting in a transport time of ~5 s. Further details have been described in the literature (Stockwell et al., 2014). All of our instruments for sampling and online measurements were placed on the platform, which can accommodate up to 1820 kg of equipment and operators. Measurements were focused on the "stack burns", for which fires lasted a few minutes up to 40 minutes.

For this study, we investigated 20 stack fires of vegetation types abundant in the western US, representing coniferous ecosystems, including ponderosa pine (PIPO), lodgepole pine (PICO), Engelmann spruce (PIEN), Douglas-fir (PSME) and subalpine fir (ABLA), with replicate burns for most of these types (Table 1). Some of the fires proceeded by burning of an individual fuel component such as litter, canopy, duff and rotten logs. Other fires simulated actual biomass in the coniferous ecosystem by mixing various fuel components in realistically recreated ecosystem matrices using the first order fire effects model (FOFEM) (Reinhardt et al., 1997).

### 2.2 Instrumentation

#### 2.2.1 Collection of HONO, NO$_x$ and nitrate for isotopic analysis

HONO was completely collected for isotopic analysis using an annular denuder system (ADS) (Chai and Hastings, 2018). The ADS system deployed in this laboratory experiment consisted of a Teflon particulate filter, a Nylasorb filter to remove HNO$_3$, followed by two annular denuders, each coated with a solution of 10 mL of Na$_2$CO$_3$ (1% w/v) + glycerol (1% v/v) + methanol−H$_2$O solution (1:1 volume ratio) following a standard EPA method. Methanol and glycerol are certified ACS plus with a purity of ≥99.8% and ≥99.5%, respectively. After coating, the denuders are dried using zero air and capped immediately. Within 6 hours after each collection, the coating was extracted in 10 mL of ultrapure water (18.2 MΩ) in two sequential 5 mL extractions. The extracted solution with a pH of ~10 was transported to Brown University for concentration and isotopic analysis 3-14 days after the sampling. The timescales for sample extraction and isotopic analysis preserve both the solution concentration and isotopic composition of HONO in the form of nitrite (Chai and Hastings, 2018). The two-denuder set up allows us to minimize the interference for both concentration and isotopic analysis from other N-containing species that could be trapped and form nitrite in residual amounts on the denuders, especially NO$_2$. Our method development study showed NO$_2$ tends to absorb in the same amount (difference <4%) on the walls of each denuder in a train setup, which is consistent with other studies (Perrino et al., 1990; Zhou et al., 2018). On the basis of this validation, the second denuder extract is used to correct the first denuder extract for both concentration and isotopic composition (Chai and Hastings, 2018). Note HONO levels



were above the minimum detection limit (0.07 uM) and the breakthrough amount of
HONO threshold is far from being reached given the concentrations (Table 1), flow rate
(~ 4 L/min) and collection times (5 - 40 min). The necessary amount minimum of nitrite
collected for isotopic analysis is 10 - 20 nmol.
Following the ADS, to avoid scrubbing of HONO, are a flow meter (Omega) and the
$NO_x$ collection system for analysis of $\delta^{15}N\text{-}NO_x$ (Fibiger et al., 2014; Fibiger and
Hastings, 2016; Wojtal et al., 2016). In brief, $NO_x$ is collected in a solution containing
0.25 M $KMnO_4$ and 0.5 M NaOH which oxidizes NO and $NO_2$ to $NO_3^-$ by pumping
sampled air through a gas washing bottle with a 65 Watts diaphragm vacuum pump. The
flow rate (~4L/min with 1% uncertainty) is controlled with a critical orifice inserted
between the pump and gas stream outlet, and is monitored and recorded with a flow
meter placed prior to the $NO_x$ collector. The $NO_x$ trapping solution blanks are also
collected every day to quantify background $NO_3^-$ for concentration and isotopic blank
corrections. The Omega flow meter was calibrated with another flow meter (Dry Cal Pro)
by varying flow rates. Within a day after collection, we stabilized the samples in the wet
chemistry lab in the Fire Science Lab by adding 30% w/w $H_2O_2$ that reduces $MnO_4^-$ to
$MnO_2$ precipitate before being shipped back to Brown University for further processing.
This effectively excludes the possible interferences from $NH_3$ that could be oxidized to
$NO_3^-$ by $MnO_4^-$ after a week (Miller et al., 2017) and references therein). The samples
were neutralized with 12.1 N HCl in the Brown lab, before concentration measurement
and isotopic analyses. $NO_3^-$ on the upstream Millipore filters and $HNO_3$ from the
Nylasorb filters, if there was any, were extracted by sonicating the filters in ~30 mL
ultrapure $H_2O$ (18.2 MΩ). Samples with $[NO_3^-] > 1$ μM were analyzed for isotopic
composition (concentration techniques detailed below).
All treated samples from both HONO collection and $NO_x$ collection and their
corresponding blanks were analyzed offline for concentrations of $NO_2^-$ and $NO_3^-$ with a
WestCo SmartChem 200 Discrete Analyzer colorimetric system. The reproducibility of
the concentration measurement was ±0.3 μmol $L^{-1}$ (1σ) for $NO_2^-$ and ±0.4 μmol $L^{-1}$ for
$NO_3^-$ when a sample was repeatedly measured (n = 30). A detection limit of 0.07 μmol
$L^{-1}$ for $NO_2^-$ and 0.1 μmol $L^{-1}$ for $NO_3^-$ was determined, and no detectable nitrite or
nitrate was found in the blank denuder coating solution, whereas blank $NO_3^-$
concentrations of ~5 μM are typical for the $NO_x$ collection method (Fibiger et al., 2014;
Wojtal et al., 2016). Note that $NO_3^-$ concentration was measured on the ADS solutions to
verify whether and to what extent $NO_2^-$ was oxidized to $NO_3^-$ on denuder walls because
the denitrifier method will convert both $NO_3^-$ and $NO_2^-$ to $N_2O$ for isotopic analysis (see
below). In addition, samples collected with a mist chamber/ion chromatography system
(described in Sect. 2.2.2) were also tested for their concentrations and only those with
sufficient nitrite quantity were further analyzed for isotopic composition.
**225 2.2.2 $NO_x$ and HONO online concentration measurement**
NO and $NO_x$ concentrations were measured with a Thermo Scientific Model 42i
chemiluminescence NO/$NO_x$ analyzer, with ±0.4 ppbv precision and 0.2 ppbv zero noise
at 1 minute time resolution. In the NO channel, $O_3$ generated by an ozonator titrates NO
to excited state $NO_2$ which subsequently produces luminescence that is proportional to
NO concentration. In the $NO_x$ channel, the sample gas stream first flows through a heated
molybdenum catalyst (325 °C) that converts $NO_2$ to NO before entering the $NO+O_3$
reaction chamber. The auto cycle mode ($NO/NO_x$) switches the mode solenoid valve
automatically on a 10 second cycle so that NO, $NO_2$, and $NO_x$ concentrations are
determined. It is known that some $NO_y$ species including HONO, $HNO_3$, organic nitrate
and PAN can be partially converted to NO in the hot molybdenum catalyst, causing
positive artifacts in measured $NO_x$ (Reed et al., 2016). In this study, only the HONO
interference was corrected using our ADS measured HONO concentration; contributions
from $HNO_3$, PAN and gaseous organic nitrate are not of major concern because no photo-
oxidation is involved in indoor fires (Koss et al., 2018; Selimovic et al., 2018; Stockwell
et al., 2014). In addition, we do not expect that other reactive nitrogen species such as
$NH_3$ and hydrogen cyanide (HCN) interfere with $NO_2$ measurement. A particulate matter
filter (Millipore, 1μm PTFE) was always placed before the inlet of the $NO_x$ analyzer. The
NO channel was calibrated before and after the entire Fire Lab experiments with standard
NO (10 ppmv in $N_2$) diluted with zero air (Thermo Fisher Scientific, Model 111) via a
gas dilution calibrator (Thermo Fisher Scientific, Model 146i) and $NO_2$ response of the
$NO_x$ channel using $O_3$ titration is within ±5% accuracy. The $NO_x$ measurement verified
the concentration of the $NO_x$ collected for isotopic analysis, and the original $NO_x$ data is
available in the NOAA FIREX archive (FIREX, 2016).
HONO and $HNO_3$ concentrations were measured using the University of New
Hampshire's dual mist chamber/ion chromatograph system (Scheuer et al., 2003) with the
sampling inlet placed right next to that of the ADS. The dual channel IC system is custom
built using primarily Dionex analytical components. Briefly, automated syringe pumps
are used to move samples and standard solutions in a closed system, which minimizes
potential contamination. A concentrator column and 5 ml injections were used to improve
sensitivity. Eluents are purged and maintained under a pressurized helium atmosphere.
Background signal is minimized using electronic suppression (Dionex-ASRS). The
chromatography columns and detectors are maintained at 40 °C to minimize baseline
drifting. A tri-fluoro-acetate tracer spike into the ultra-clean sampling water is used as an
internal tracer of sample solution volume, which can decrease due to evaporation in the
exhaust flow by 10-20% depending on the ambient conditions and length of the sample
integration interval. In this work, each sample integrated 5 minutes of collection. The
spike was analyzed to correct the final mist chamber sampled solution volume with an
uncertainty of ±3%. This system has been deployed to various field studies for HONO
measurement (Dibb et al., 2002; Stutz et al., 2010) and showed reasonable
intercomparison with other HONO measurement techniques (within 16% uncertainty)
during the 2009 SHARP campaign in Houston (Pinto et al., 2014). The detection limits
for $HNO_3$ and HONO are 10 ppt for 5-minute sample integrations. During the
experiments, two mist chambers were operated to collect gas samples in parallel, each
with an integration interval of 5 min. One channel of the IC was utilized for concentration
measurement; in the other, the mist chamber's solution was transferred into a sample
bottle using the syringe pump, and the collected solution was brought to Brown
University for isotopic analysis of $HNO_3$ if sufficient amount (10-20 nmol) was collected
for each sample.

In addition to MC/IC, the HONO mixing ratios were also measured using high time-





resolution (~1 second) measurement techniques including open-path Fourier transform
infrared spectroscopy (OP-FTIR) (Selimovic et al., 2018), cavity enhanced spectrometer
(CES) (Min et al., 2016; Zarzana et al., 2018), and proton-transfer-reaction time-of-flight
mass spectrometer (PTR-ToF). Inlet ports of CES and PTR-ToF were placed 5' apart
from, but at the same height on the platform as those for ADS and MC/IC, while the OP-
FTIR had an open path cell at the stack. The smoke has been shown to be well-mixed at
the sampling platform (Christian et al., 2004) and the mean HONO mixing ratios across
each fire obtained from the four techniques were compared with that retrieved from ADS
collection. This offers comprehensive verification of complete capture of HONO by ADS
that is extremely important for conserving the isotopic composition of HONO.
The details of OP-FTIR are described in previous works (Selimovic et al., 2018;
Stockwell et al., 2014). The setup included a Bruker MATRIX-M IR cube spectrometer
with a mercury cadmium telluride (MCT) liquid-nitrogen-cooled detector interfaced with
a 1.6 m base open-path White cell. The white cell was positioned on the platform and its
open path spanned the width of the stack. This facilitates direct measurement across the
rising emissions. The optical path length was set to 58 m. The IR spectra resolution was
0.67 cm$^{-1}$ from 600–4000 cm$^{-1}$. Pressure and temperature were continuously recorded
with a pressure transducer and two temperature sensors respectively, which were placed
adjacent to the White cell optical path. They were used for spectral analysis. Time
resolution for stack burns was approximately 1.37 s. The OP-FTIR measures $CO_2$, CO,
$CH_4$, a series of volatile organic compounds and various reactive nitrogen species
(Selimovic et al., 2018).–Mixing ratios of HONO were retrieved via multicomponent
fitting to a section of the mid-IR transmission spectra with a synthetic calibration
nonlinear least-squares method (Griffith, 1996; Yokelson et al., 2007), and both the
HITRAN spectral database and reference spectra recorded at the Pacific Northwest
National Laboratory (Rothman et al., 2009; Sharpe et al., 2004; Johnson et al., 2010,
2013) were used for the fitting. The uncertainty is ~10% for the HONO mixing ratio
measurement and the detection limit is no more than a few ppb as reported in previous
studies (Stockwell et al., 2014; Veres et al., 2010).
Nitrous acid measurements by cavity enhanced spectroscopy used the airborne cavity
enhanced spectrometer, ACES, recently described by Min *et al.* (2016). This instrument
consists of two channels, one measuring over the spectral range from 438-468 nm where
glyoxal (CHOCHO) and $NO_2$ have structured absorption bands, and one measuring over
from 361-389 nm, where HONO has structured absorption.   In the HONO channel, light
from an LED centered at 368 nm and with an output power of 450 mW and collimated
with an off-axis parabolic collector illuminates the input mirror of a 48 cm optical cavity
formed from mirrors with a maximum reflectivity R = 99.98% at 375 nm.  The effective
path length within the optical cavity is > 3 km over the region of greatest reflectivity. The
mirror reflectivity (effective path length) was calibrated from the difference in Rayleigh
scattering between He and zero air to provide an absolute calibration of the instrument
response.  A fiber optic bundle collects light exiting the optical cavity and transmits it to
a grating spectrometer with a CCD detector, where it is spectrally dispersed at a
resolution of 0.8 nm.  The resulting spectra are fit using DOASIS software (Kraus, 2006)
to determine trace gas concentrations, including $NO_2$, HONO and $O_4$.  Mixing ratios of
$NO_2$ and HONO are reported at 1 s resolution, although the $NO_2$ precision is higher in the





455 nm channel. The 1 Hz HONO precision is 800 pptv ($2\sigma$). (The precision of the
HONO instrument in ACES is somewhat degraded by the optimization of 455 nm
channel for glyoxal detection, which reduces the photon count rate on the 368 nm
channel.) The accuracy of the HONO measurement is 9%. Air was sampled directly
from stack at a height of 15 m above the fuel bed through a 1 m length of ¼" O.D. Teflon
(FEP) tubing as described in Zarzana *et al.* (2018). The residence time in the inlet and
sample cells was < 1 s. Comparison between the ACES HONO and an open path FTIR
agreed to within 13% on average, and ACES HONO was well correlated with 1Hz
measurements from a PTR-ToF ($r^2 = 0.95$) (Koss et al., 2018).
The PTR-ToF instrument used in the FIREX Fire Lab experiment is described in detail in
previous studies (Koss et al., 2018; Yuan et al., 2016). The PTR-ToF instrument is a
chemical ionization mass spectrometer typically using $H_3O$ reagent ions and a wide range
of trace gases can be detected in the range of tens to hundreds of parts per trillion (pptv)
for a 1 s measurement time. At the Fire Lab, PTR-ToF detected several inorganic species
including HONO with an uncertainty of 15%. HONO is detected at a lower sensitivity
than most trace gases in PTR-ToF, but mixing ratios for all fires were well above the
detection limit.
**2.2.3 Isotopic composition measurements**
The denitrifier method was used to complete nitrogen and oxygen isotope analyses
($^{15}N/^{14}N$, $^{18}O/^{16}O$) of $NO_3^-$ and/or $NO_2^-$, by quantitative conversion to $N_2O$ by
denitrifying bacteria *P. aureofaciens* (Casciotti et al., 2002; Sigman et al., 2001). The
isotopic composition of $N_2O$ is then determined by a Thermo Finnegan Delta V Plus
isotope ratio mass spectrometer at *m/z* 44, 45 and 46 for $^{14}N^{14}N^{16}O$, $^{14}N^{15}N^{16}O$ and
$^{14}N^{14}N^{18}O$, respectively. Sample analyses were corrected against replicate measurements
of the $NO_3^-$ isotopic reference materials USGS34, USGS35, and IAEA-NO-3 (Böhlke et
al., 2003). Precisions for $\delta^{15}N$-HONO and $\delta^{15}N$-$NO_x$ isotopic analysis across each of the
entire methods are ±0.5‰ and ±1.3‰, respectively (Chai and Hastings, 2018; Fibiger et
al., 2014). $\delta^{18}O$-$N_2O$ from the $NO_x$ collection samples was measured but is not reported
as $\delta^{18}O$-$NO_x$ because it is greatly impacted by $MnO_4^-$ oxidation and does not represent
the $\delta^{18}O$-$NO_x$ in the sample air. The total $\delta^{15}N$ of the starting biomass ($\delta^{15}N$-biomass)
was measured at the Marine Biological Laboratory Ecosystems Center Stable Isotope
Facility. The materials measured for $\delta^{15}N$-biomass (Table S1) cover most but not all the
biomass types burned in the experiments depending on availability of the leftover
materials. Analyses were conducted using a Europa ANCA-SL elemental analyzer−gas
chromatograph preparation system interfaced with a Europa 20−20 continuous-flow gas
source stable isotope ratio mass spectrometer. Analytical precision was ±0.1‰, based on
replicate analyses of international reference materials.
Collection time spanned the whole fire burning (5 min to 40 min) in order to maximize
the signal. We chose to report the samples whose concentrations are at least 30% above
the 5 μM $NO_3^-$ present in the blank $KMnO_4$ solution upon purchase (Fibiger et al., 2014),
such that the propagated error through the blank correction does not exceed the analytical
precision of ±1.5‰ for $\delta^{15}N$-$NO_x$. We found identical concentration and isotopic
signatures for both field and laboratory blanks, which ensures that no additional $NO_3^-$



contamination was introduced into the KMnO$_4$ solutions in the gas-washing bottle. In
addition, fires with high particulate loading that resulted in >50% reduction in flow rate
are not considered for isotopic analysis because the low flow rate could induce
incomplete collection with potential isotopic fractionation that might not represent BB
emissions.
**3. Results and discussion**
**3.1 Temporal evolution of HONO and HNO$_3$ from direct BB emissions**
The time series of HONO and HNO$_3$ concentrations measured by MC/IC at 5-minute
resolution for each available stack fire are shown in Fig. 1, and original data can be found
in the NOAA data archive (FIREX, 2016). HNO$_3$ concentrations were nearly two orders
of magnitude lower than typical HONO concentrations. The constant low concentration
of HNO$_3$ from fresh emissions across all fires is consistent with the findings in (Keene et
al., 2006), confirming HNO$_3$ is not a primary reactive nitrogen species in fresh smoke.
Rather, it is largely produced secondarily in aged smoke and nighttime chemistry. Both
HONO and HNO$_3$ mixing ratios reach their peak in the first five minutes, except for fire
no. 12 (Engelmann spruce - duff), from which HONO concentration remains nearly
constant over the course of each fire, but much lower than HONO concentration of the
rest of the fires. The largest HONO and HNO$_3$ were emitted from burning subalpine fir-
Fish Lake canopy (fire no. 15), integrated concentration of up to 177 ppbv and 1.9 ppbv
in the first 5-minute sample, respectively. We note that fires no. 12 has the smallest MCE
value 0.868 (FIREX, 2016). In general, the closer the MCE value is to 1, the more likely
N-oxidation (e.g. NO$_x$ and HONO) dominates over N-reduction (e.g. NH$_3$ and HCN) as a
result of flaming; when MCE approaches 0.8, more smoldering occurs such that N-
reduction becomes dominant (Ferek et al., 1998; Goode et al., 1999; McMeeking et al.,
2009; Yokelson et al., 1996, 2008). Accordingly, the smoldering combustion condition of
fire no. 12 leads to lower concentration of oxidized nitrogen species than the rest of the
fires in this study. In addition, HONO/NO$_x$ ratio ranged from 0.13 to 0.53 with a mean of
0.29±0.12, comparable with previous results of laboratory experiments (0.11±0.04) and
field experiments (0.23±0.09) (Akagi et al., 2013; Burling et al., 2010, 2011)
**3.2 Verification of ADS collected HONO concentration**
The HONO collected with the ADS represents a mean value over the course of each
entire burn. We first compare HONO concentration recovered from the ADS, denoted as
[HONO]$_{ADS}$, with that measured with the collocated MC/IC when both measurements
were available (Fig. 2). The comparison demonstrates good consistency across all fires,
with the [HONO]$_{ADS}$ of all available fires falling within the first and third quartile of
MC/IC HONO data. Additionally, we made intercomparisons between [HONO]$_{ADS}$ with
mean values of various high resolution methods including MC/IC, OP-FTIR, ACES and
PTR-ToF that are also available from the NOAA data archive (Fig. 3; FIREX, 2016). The
mean values used for the comparison are shown in Table S2. The linear regression results
for all four comparisons are:
$[HONO]_{ADS} = (1.07 \pm 0.24) [HONO]_{MCIC} - 0.72$  Eq. (1)
$(R^2 = 0.63; p_{slope} < 0.001, p_{intercept}=0.95)$;

$[HONO]_{ADS} = (1.07\pm0.08) [HONO]_{ACES} - 4.63$      Eq. (2)
$(R^2 = 0.95; p_{slope} < 1\times10^{-6}, p_{intercept}=0.32)$;

$[HONO]_{ADS} = (1.07\pm0.22) [HONO]_{FTIR} + 5.48$      Eq. (3)
$(R^2 = 0.75; p_{slope} < 0.005, p_{intercept}=0.48)$;

$[HONO]_{ADS} = (1.08\pm0.19) [HONO]_{PTR\text{-}ToF} - 8.81$      Eq. (4)
$(R^2 = 0.87; p_{slope} < 0.005, p_{intercept}=0.28)$.

We found significant linear correlation between each of the [HONO] techniques and
[HONO] $_{ADS}$ with a slope of ~1. Note that the y-intercepts of Eq. (1)—(4) are much
smaller than the overall range of measured [HONO] (up to 121 ppbv). In addition, p-
values of the intercepts for all 4 fittings are much greater than 0.05, suggesting the
intercepts are not significantly different from zero. All data except one fall within 95%
confidence interval bounds of the overall fitting (Fig. 3). Therefore, we conclude that the
ADS method has high capture efficiency of HONO in the biomass combustion
environment, which assures the accuracy of the isotopic composition analysis and
applicability of this method for field-based biomass combustion research.

**3.3 Isotopic composition of HONO and $NO_x$ from burning different biomass**

$\delta^{15}N$ of $NO_x$ and HONO emitted from burning various biomass types in this study ranged
from -4.3 ‰ − +7.0‰ and -5.3 − +5.8‰, respectively (Table 1). There is no direct
dependence of $\delta^{15}N$ on concentration of either HONO or $NO_x$ (Figure S1). In Fig. 4, $\delta^{15}N$
values of $NO_x$ and HONO are shown for each biomass type. Each value represents a
mean weighted by concentration (if multiple samples were collected for a biomass type)
with error bars representing propagation of replicate variation and method precision. For
biomass types burned in replicate (PIPO, PICO, PIEN, PSME), the $\delta^{15}N$-$NO_x$ and $\delta^{15}N$-
HONO variation within a given biomass type is smaller than the full range across all fuel
types. Additionally, we note that the variations of $\delta^{15}N$-$NO_x$ and $\delta^{15}N$-HONO for PIPO
and $\delta^{15}N$-HONO for PIEN are larger than the method analytical precision of $\delta^{15}N$-$NO_x$
(1.5‰) and $\delta^{15}N$-HONO (0.5‰), respectively, which represents fire-by-fire variation
likely due to different combustion conditions and/or different fuel compositions. For
example, fuel moisture content derived from the original biomass weight and dry biomass
weight reveal that the PIPO burned in fire no.3 had more moisture content (48.1%) than
fire no.2 (32.1%), which could affect combustion temperature and thus product
formation. Fig. 4 also illustrates burning different biomass parts from specific vegetation
can result in fairly diverse $\delta^{15}N$-HONO and $\delta^{15}N$-$NO_x$, e.g. among PIPO mixture, canopy
and litter, as well as between PIEN mixture and duff.

Our $\delta^{15}N$-$NO_x$ range falls well within the range (-7‰ − +12‰) found in the FLAME-4
experiment (Fibiger and Hastings, 2016). The FLAME-4 study investigated $NO_x$
emissions from burning a relatively large range of vegetation biomass from all over the
world, and found a linear relationship (Eq. (5)), indicating that 83% of the variation of



$\delta^{15}$N-NO$_x$ is explained by $\delta^{15}$N-biomass. The biomass types burned in this work focused on vegetation in the western U.S., and differ greatly from that in FLAME-4, with Ponderosa pine being the only common biomass between the two studies. Specifically, the $\delta^{15}$N-biomass range (-4.2‰ – +0.9‰) for this work is much narrower than that of the FLAME-4 experiment (-8‰ – +8‰).

$$\delta^{15}\text{N-NO}_x = 0.41\ \delta^{15}\text{N-biomass} + 1.0\ (r^2=0.83,\ p<0.001) \qquad \text{Eq. (5)}$$

To compare with the relationship found in Fibiger and Hastings (2016), we mass weighted the contributions from different components of the same biomass type. For the same type of biomass, $\delta^{15}$N-biomass varies amongst different parts of the vegetation with differences as great as 4.1‰, 2.4‰, 4.6‰ and 2.6‰ for PIPO, PICO, PSME and PIEN, respectively (Table S1). In the FIREX experiments, many of the burns were conducted for mixtures of various vegetation parts. For instance, one PIPO fire contains canopy (~30%), litter (~28%), and other parts (~42%) including duff and shrub, and the compositions vary slightly amongst each burn. Therefore, the $\delta^{15}$N of a particular biomass mixture is mass weighted according to its composition contribution from each part (Table S1). Similarly, the $\delta^{15}$N-NO$_x$ and $\delta^{15}$N-HONO from fires of different biomass parts are weighted by concentrations for each biomass type, i.e. PIPO (including mixture, canopy and litter) and PIEN (including mixture and duff), to produce a signature associated with combustion of that biomass type.

For purpose of comparison among different biomass types, we average $\delta^{15}$N-NO$_x$ ($\delta^{15}$N-HONO) weighted by concentrations for each biomass type, i.e. PIPO (including mixture, canopy and litter) and PIEN (including mixture and duff) (all data are listed in Table S3). Linear regressions between $\delta^{15}$N-HONO and $\delta^{15}$N-biomass, as well as that between $\delta^{15}$N-NO$_x$ and $\delta^{15}$N-biomass, show that both $\delta^{15}$N-HONO and $\delta^{15}$N-NO$_x$ increase with $\delta^{15}$N-biomass in general (Fig. S2). However, the linear regressions performed here are limited by small datasets (4 data points each) and unsurprisingly yield insignificant linear correlations for $\delta^{15}$N-HONO (or $\delta^{15}$N-NO$_x$) versus $\delta^{15}$N-biomass (p values are 0.1 and 0.5, respectively). Still, combining our results of $\delta^{15}$N-NO$_x$ versus $\delta^{15}$N-biomass from this work with those from the FLAME-4 study (Fibiger and Hastings, 2016) results in a significant linear correlation (Eq. (6)) and is shown in Fig. 5. Despite differences in burned biomass types between the two studies, our $\delta^{15}$N-NO$_x$ reasonably overlap with the FLAME-4 results within our $\delta^{15}$N-biomass range. The relationship between $\delta^{15}$N-NO$_x$ and $\delta^{15}$N-biomass (Eq. (6)) for the combined data highly reproduces that obtained solely from FLAME-4 study (Eq. (5)) and constrains the dependence of $\delta^{15}$N-NO$_x$ on $\delta^{15}$N-biomass.

$$\delta^{15}\text{N-NO}_x = (0.42\pm0.17)\ \delta^{15}\text{N-biomass} + 1.3\ (r^2=0.71,\ p<0.001) \qquad \text{Eq. (6)}$$

The mean values weighted by concentration plotted in Fig. 4 show $^{15}$N of HONO is consistently slightly more depleted than that of NO$_x$ ($\delta^{15}$N-HONO < $\delta^{15}$N-NO$_x$) across all the biomass types, except for PIPO-litter which results in an opposite relationship between $\delta^{15}$N-HONO and $\delta^{15}$N-NO$_x$. Furthermore, $\delta^{15}$N-HONO is linearly correlated with $\delta^{15}$N-NO$_x$ following a relationship of Eq. (7) within the $\delta^{15}$N-NO$_x$ and $\delta^{15}$N-HONO



range obtained in the current study (Fig. 6). This provides potential insights into HONO-
$NO_x$ interactions and HONO formation pathways in fresh emissions from biomass
burning. Although a number of studies on wildfire biomass burning have suggested that
partitioning of N emissions between $NO_x$ and $NH_3$ depends on combustion conditions
represented by MCE (Ferek et al., 1998; Goode et al., 1999; McMeeking et al., 2009;
Yokelson et al., 1996, 2008), HONO formation pathways remain unclear (Alvarado et al.,
2009, 2015; Nie et al., 2015).
$\delta^{15}$N-HONO = 1.01 $\delta^{15}$N-$NO_x$ - 1.52 ($R^2$ = 0.89, p<0.001)                Eq. (7)
Previous mechanistic studies on combustion of biomass/biofuel model compounds in a
well controlled closed system have investigated detailed nitrogen chemistry in the gas
phase, suggesting $NO_x$ and HONO are formed from chain reactions involving oxidation
of precursors $NH_3$ and HCN, which are produced via devolatilization and pyrolysis of
amines and proteins in biomass/biofuel (Houshfar et al., 2012; Lucassen et al., 2011).
When the combustion conditions favor the oxidation of $NH_3$ and HCN, NO is first
formed and the chain reactions control the cycling of reactive nitrogen species (NO, $NO_2$
and HONO). Detailed and mechanistic nitrogen chemistry for the chemical relationship
between $NO_x$ and HONO in the combustion environment have been discussed in earlier
works (Chai and Goldsmith, 2017; Shrestha et al., 2018; Skreiberg et al., 2004). In
addition, Houshfar et al. (2012) performed biomass combustion kinetic modeling with
reduced mechanism via sensitivity analysis. From these works, we extract major
pathways (R1-R11) that are likely responsible for fast gas-phase inter-conversion
between $NO_x$ and HONO within the combustion system. They found whether HONO is
converted from NO or $NO_2$ in series during nitrogen transformation (referred to as
nitrogen flow) critically depends on temperature. Specifically, within 1 second of
residence time, at moderate temperatures (e.g. 700 °C), dominant nitrogen flow following
NO formation in biomass combustion is NO→$NO_2$→HONO→NO, and major reactions
involving $NO_x$-HONO conversion are listed in R1-R6; at high temperatures (e.g. 850 °C
and above), the nitrogen flow cycle NO→HONO→$NO_2$→NO becomes dominant, and
major reactions involving $NO_x$-HONO are R7-R11.
$NO_2$ + HNOH → HONO + HNO                R1
$NO_2$ + HNO → HONO + NO                R2
$NO_2$ + $HO_2$ → HONO + $O_2$                R3
$NO_2$ + $H_2$ → HONO + H                R4
$NO_2$ + $C_xH_y$ (hydrocarbon) → HONO + $C_xH_{y-1}$                R5
HONO → OH + NO                R6
OH + NO → HONO                R7
HONO + $NH_2$ → $NO_2$ + $NH_3$                R8





HONO + NH → NO$_2$ + NH$_2$                                                   R9
HONO + O → NO$_2$ + OH                                                        R10
HONO + OH → NO$_2$ + H$_2$O                                                   R11
Although our studied fuels are more complicated in composition than a model system
involving no more than a few starting species, results from the above studies provide
fundamental underpinnings for biomass combustion. Also note that heterogeneous
chemistry after these species were emitted was not considered here as the residence time
of the fresh plume in our study was ~5 seconds, which is of the same magnitude as that
predicted in the nitrogen flow analysis (Houshfar et al., 2012). Kinetic isotope effects
(KIE) of these reactions have not been characterized; so only a semi-quantitative
prediction is presented here. At low temperatures, R1-R5 are all H-abstraction reactions
involving loose transition states that have significant activation energy; a primary KIE is
expected for such conditions and leads to $^{15}$N depletion in the product (HONO) (Chai et
al., 2014; Matsson and Westaway, 1999, and references therein). Additionally, R6 is a
unimolecular dissociation reaction with no reaction barrier, and hence R6 could be
expected to have a small kinetic isotope effect enriching $^{15}$N in HONO, somewhat
offsetting the depletion that arose from R1-R5. Consequently, the overall isotope effect of
R1-R6 would lead to $\delta^{15}$N-HONO < $\delta^{15}$N-NO$_x$ by a small difference, consistent with our
results (Fig. 4). On the other hand, the KIE for the reactions R7-R11 at higher
temperatures (> 850 °C) is expected to enrich $^{15}$N in HONO relative to NO$_x$ (Chai and
Dibble, 2014), leading to an opposite isotope effect to that predicted at lower
temperatures.
Temperatures of the biomass combustion process span a large range involving different
processes including preheating, drying, distillation, pyrolysis, gasification (aka "glowing
combustion") and oxidation in turbulent diffusion flames at a range of temperatures
associated with changing flame dynamics (Yokelson et al., 1996). Despite this
complexity, our measured slight $^{15}$N enrichment in NO$_x$ compared to HONO (Table 1,
Fig. 4) suggests that the reactions R1-R6 play a more important role than R7-R11 in
HONO formation during the FIREX Fire Lab experiments.
**3.4 Isotopic composition of nitrates collected on particle filters**
All Nylasorb filter extract solutions showed no detectable NO$_3^-$ and NO$_2^-$ concentrations,
indicating no significant amount of HNO$_3$ was collected on these filters, which is
consistent with the very low concentrations measured by MC/IC (note that low
concentration and limited sample volume also preclude further isotopic analysis of HNO$_3$
collected by MC/IC). By contrast, we found 5 out of 20 particulate filter extract solutions
had detectable NO$_3^-$ concentration that were sufficient (10 nmol N) for isotopic
composition analysis (Table 1). $\delta^{15}$N and $\delta^{18}$O reported here are considered to represent
NO$_3^-$(p). $\delta^{15}$N-NO$_3^-$(p) of the five samples (burns) range from -10.6 to -7.4 ‰, all of
which are more $^{15}$N depleted than that of HONO and NO$_x$. In addition, the smaller range
of $\delta^{15}$N-NO$_3^-$ than that of $\delta^{15}$N-HONO and $\delta^{15}$N-NO$_x$ rules out possible transformation of
$NO_x$ and HONO to nitrate on the filters, which could distort the isotopic composition of
$NO_x$ and HONO.
In the FLAME-4 experiments (Fibiger and Hastings, 2016), only one particulate filter had
captured $NO_3^-$(p) above the concentration detection limit, whereas $HNO_3$ collected on
Nylasorb filters from 7 experiments were above the concentration detection limit and
therefore only $\delta^{15}N$-$HNO_3$ (-0.3 – 11.2‰) were reported. The contrast with our filter
results are likely attributed to different formation mechanisms under different conditions,
in addition to variation of fuel types. Of the 7 detectable $HNO_3$ collections from FLAME-
4, 5 represented room burns for which samples were collected from smoke aged for 1-2
hours in the lab, and the sampled $HNO_3$ was likely a secondary product. By contrast all
our observed $NO_3^-$(p) were in fresh emissions and may have been derived from plant
nitrate (Cárdenas-Navarro et al., 1999) and/or combustion reactions. There have been no
other studies on $\delta^{15}N$ of $NO_3^-$(p) and $HNO_3$ directly emitted from fresh plumes to the best
of our knowledge, so more investigation using both laboratory work (isotope effect) and
kinetic modeling will be needed in order to understand formation mechanisms of $HNO_3$
and $NO_3^-$(p) in the biomass combustion process and their respective isotope effects.
In addition to $\delta^{15}N$, we report $\delta^{18}O$ of HONO and $NO_3^-$(p) directly emitted from biomass
burning plumes with ranges of 5.2‰ – 15.2‰ and 11.5‰ – 14.8‰, respectively. These
are the first observations reported for $\delta^{18}O$ of reactive nitrogen species directly emitted
from biomass burning and low values are expected for the $\delta^{18}O$, which, in this case, is
mainly extracted from that of molecular oxygen ($\delta^{18}O$ = ~23.5‰) (Kroopnick and Craig,
1972), biomass/cellulose ($\delta^{18}O$ = 15‰–35‰), and/or biomass contained water ($\delta^{18}O$ = ~
0‰– 16‰) (Keel et al., 2016). In field studies where photochemistry and $O_3$ are
inevitably involved in the reactive nitrogen cycle in various stages of aged plumes, we
expect to see much more elevated $\delta^{18}O$ values of HONO and $NO_3^-$(p) due to the
extremely high value of $\delta^{18}O$-$O_3$ (~110‰) (Vicars and Savarino, 2014). Therefore, the
$\delta^{18}O$ found in the lab is helpful in understanding conditions where photochemistry would
not apply (e.g. nighttime fresh smoke) and should be distinguishable from the expected
higher $\delta^{18}O$ that would be found in aged smoke and/or daytime fresh smoke.
**4 Conclusions**
In this study we applied new methods for characterizing the isotopic composition of
reactive nitrogen species including $NO_x$ ($\delta^{15}N$), HONO ($\delta^{15}N$ and $\delta^{18}O$), and $NO_3^-$(p)
($\delta^{15}N$ and $\delta^{18}O$) emitted directly from biomass burning. We measured fresh (stack)
emissions from 20 laboratory fires of different fuels during the 2016 FIREX Fire Lab
experiments. $NO_x$, HONO and $HNO_3$ emitted in fresh smoke reached their peak in most
of our fires within five minutes of ignition of biomass (i.e. when flaming combustion
peaked). The HONO mixing ratio was typically ~2 orders of magnitude larger than
$HNO_3$, and HONO/$NO_x$ ratio ranged from 0.13 to 0.53.
Our HONO collection method (ADS) for isotopic analysis was applied to biomass
burning (BB) for the first time. The good agreement for concentration comparison
between our method and 4 high time-resolution HONO concentration methods suggests



high collection efficiency of HONO from BB emissions, which ensures accurate isotopic
compositional analysis. Comparison with concurrent observations and a previous study
show that the combination of our HONO and $NO_x$ collection methods are compatible,
allowing for simultaneous determination of the isotopic composition of both HONO and
$NO_x$. This provides important potential for investigating the photochemical and non-
photochemical relationships between HONO and $NO_x$ in a variety of environments, and
especially in BB plumes.

$\delta^{15}N$-$NO_x$ emitted from burning various Western U.S. biomass types in this study ranged
from -4.3 ‰ to +7.0 ‰, falling well within the range found by (Fibiger and Hastings,
2016), although the vegetation types were much broader in the earlier study. We report
the first $\delta^{15}N$-HONO emitted directly from burning, ranging from -5.3 to +5.8‰. $\delta^{15}N$-
$NO_x$ and $\delta^{15}N$-HONO range derived from BB can be further compared with that from
other sources using the same methods presented here, and provide insights into source
signatures for both $NO_x$ and HONO. This study also showed the important capability of
determining $\delta^{18}O$-HONO and $\delta^{18}O$-$NO_3^-$(p) from BB plumes, and we expect $\delta^{18}O$ of both
HONO and $NO_3^-$(p) produced under photochemical conditions will be much higher than
the lab results due to the important role of $O_3$ in reactive nitrogen oxidation.

Interestingly, the linear correlation between $\delta^{15}N$-HONO and $\delta^{15}N$-$NO_x$ for the biomass
we studied suggests systematic co-produced $NO_x$ and HONO occurs in the combustion
and both of them are released as primary pollutants in fresh smoke. The relationship
between $\delta^{15}N$-HONO and $\delta^{15}N$-$NO_x$ likely reflects the gas phase biomass combustion
process for flow of reactive nitrogen species at moderate combustion temperatures (< 800
°C). This correlation is potentially useful to distinguish HONO sources and formation
pathways in the environment. Determining these relationships in real wildfire smoke will
be essential for better constraint on $NO_x$ and HONO budgets, and eventually may
improve ozone and secondary aerosol predictions for regional air quality.


*Data availability.* The data from the laboratory tests are available on request from the corresponding authors. Data from the 2016 Missoula Fire lab are available here: https://esrl.noaa.gov/csd/groups/csd7/measurements/2016firex/FireLab/DataDownload/

*Supplement.*

*Author contribution.* JC, MH and JD designed this work. JC and DJM conducted the sample collections at the Fire Lab, with additional support from MH, JD and ES. JC carried out the isotopic composition measurements; DJM supported the isotopic research and interpretation. ES helped analyze the MC/IC data. VS and RY provided the OP-FTIR data. KJZ and SSB provided the ACES data. ARK and CW provided the PTR-ToF data. JC wrote the manuscript, and all authors provided edits and feedback.

*Competing interests.* The authors declare that they have no conflicts of interest.

*Acknowledgement.* This work was supported by funding from the National Oceanic and Atmospheric Administration (AC4 Award NA16OAR4310098 to MH) and the National Science Foundation (AGS-1351932 to MH). The FIREX Fire Lab study was supported in part by the NOAA Climate Office's Atmospheric Chemistry, Carbon Cycle, and Climate program. We are grateful for Ruby Ho for laboratory support and Marshall Otter for the biomass $\delta^{15}$N analysis. We also thank James Roberts and Matthew Coggon for helpful discussions.

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





Table 1. Concentration (mean, derived from solution concentration and flow rate) and N isotopic composition for various biomass burning experiments, unit for all δ denotations is ‰. MCE values are extracted from NOAA FIREX fire archive. Note: fire no. 1, 7 and 13 were missing due to technical issues; $NO_x$ results are only shown when blank/sample ratio is <70%. Biomass acronyms are defined in Sect. 2.1; in addition, d—duff, c—canopy, l—litter.

| Biomass | Fire no. | HONO (ppbv) | $\delta^{15}N$-HONO | $\delta^{18}O$-HONO | $NO_x$ (ppbv) | $\delta^{15}N$-$NO_x$ | $\delta^{15}N$-biomass | $\delta^{15}N$-p-$NO_3^-$ | $\delta^{18}O$-p-$NO_3^-$ | HONO/$NO_x$ | MCE |
|---|---|---|---|---|---|---|---|---|---|---|---|
| PIPO | 2 | 19.9 | -5.3 | 12.6 | 147.9 | -1.1 | 0.3 | -7.5 | 14.3 | 0.13 | 0.93 |
| PIPO | 3 | 35.8 | 1.7 | 11.6 | 124.7 | 2.3 | 0.3 | | | 0.29 | 0.94 |
| PIPO | 4 | 152.9 | -3.1 | 10.6 | 716.8 | -3.6 | 0.3 | | | 0.21 | 0.93 |
| PICO | 5 | 74.8 | -2.3 | 8.8 | 170.8 | -1.1 | -3.4 | -7.4 | 14.8 | 0.44 | 0.93 |
| PICO | 6 | 17.6 | -1.9 | 8.4 | 94.7 | 1.4 | -3.4 | | | 0.19 | 0.94 |
| PIEN | 8 | 25.7 | -1.7 | 14.6 | 91.7 | 0.1 | -2.4 | | | 0.28 | 0.92 |
| PIEN | 9 | 21.3 | -4.8 | 9.5 | 73.6 | -1.3 | -2.8 | | | 0.29 | 0.93 |
| PSME | 10 | 42.2 | -0.5 | 5.2 | 229.7 | 1.9 | -1.4 | -10.6 | 14.5 | 0.18 | 0.94 |
| PSME | 11 | 112.3 | -0.4 | 15.2 | 571.8 | 3.3 | -2.0 | | | 0.20 | 0.95 |
| PIEN-d | 12 | 17.1 | -4.6 | 8.5 | 36.2 | -4.3 | -1.4 | -9.9 | 11.5 | 0.47 | 0.87 |
| PSME | 14 | 25.3 | 0.1 | 14.9 | 70.0 | 2.1 | -1.9 | | | 0.36 | 0.93 |
| ABLA-c | 15 | 51.0 | 2.1 | 9.9 | 95.5 | 3.4 | -2.6 | -8.9 | 12.7 | 0.53 | 0.89 |
| PIPO-l | 16 | 70.0 | 5.8 | 7.5 | 443.3 | 5.2 | 0.9 | | | 0.16 | 0.95 |
| PIEN-c | 17 | 47.1 | 6.1 | 14.8 | | | -3.5 | | | | 0.89 |
| PSME-c | 18 | 45.3 | 2.5 | 14.0 | | | -1.4 | | | | 0.93 |
| PIPO-c | 19 | 23.8 | 5.3 | 14.8 | 73.3 | 7.0 | -0.1 | | | 0.32 | 0.93 |
| PICO-c | 20 | 52.5 | 3.0 | 14.9 | | | -3.1 | | | | 0.94 |
| PICO-l | 21 | 9.9 | 0.3 | 15.2 | | | -4.2 | | | | 0.93 |
| PSME-l | 22 | 40.0 | 1.9 | 10.2 | | | -2.3 | | | | 0.95 |
| ABLA-c | 23 | 40.8 | 0.5 | 12.2 | | | -2.6 | | | | 0.95 |

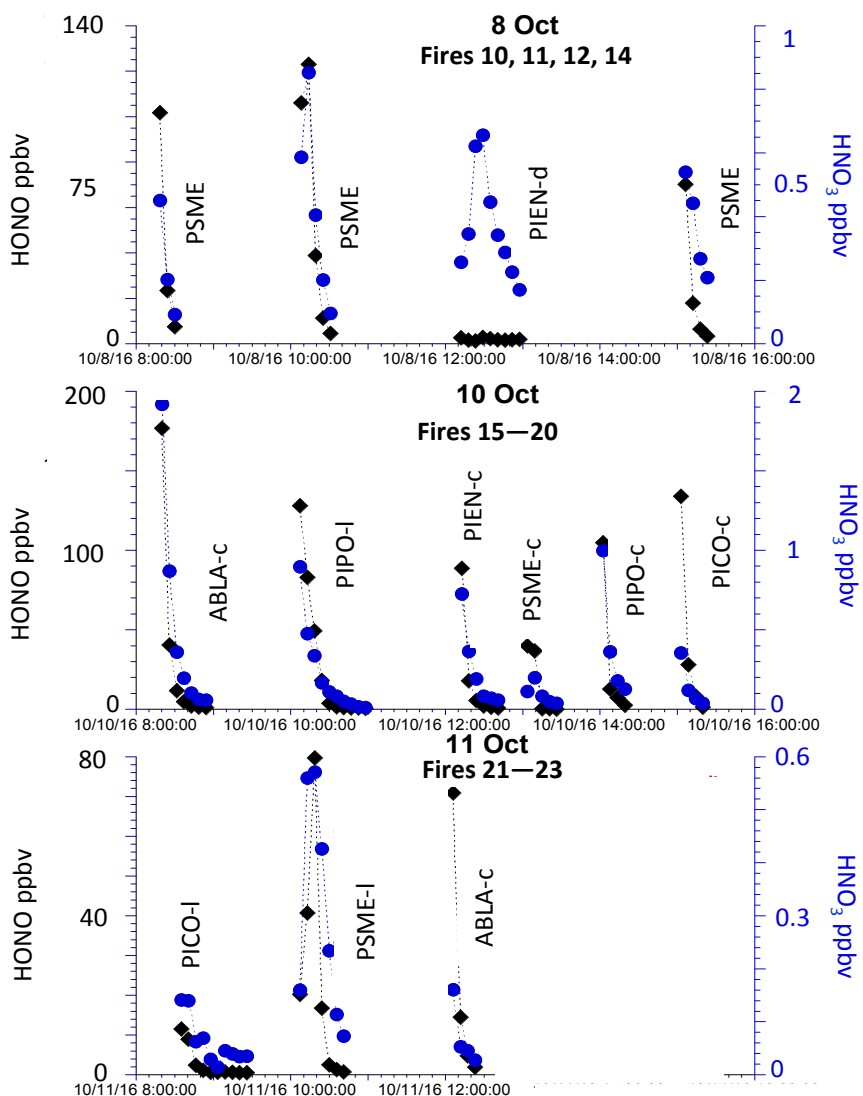

**Figure 1.** Temporal profile of HONO (black diamond) and $HNO_3$ (blue circle) concentration for various stack fires (fire numbers are referred to Table 1).

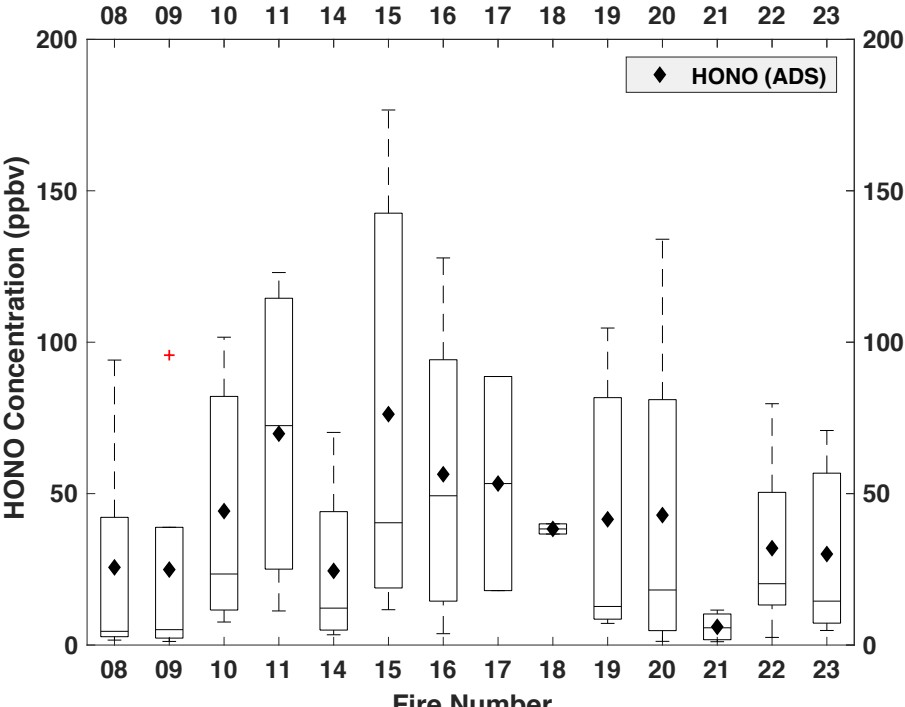

**Figure 2.** Box plot of MC/IC HONO measurement with 5 minutes resolution over the course of each fire. Each box whisker represents 5th, 25th, 50th, 75th, 95th percentile of HONO concentration during each collection period. Black diamond is the mean HONO concentration recovered from ADS collection. The red cross symbolize outliers.

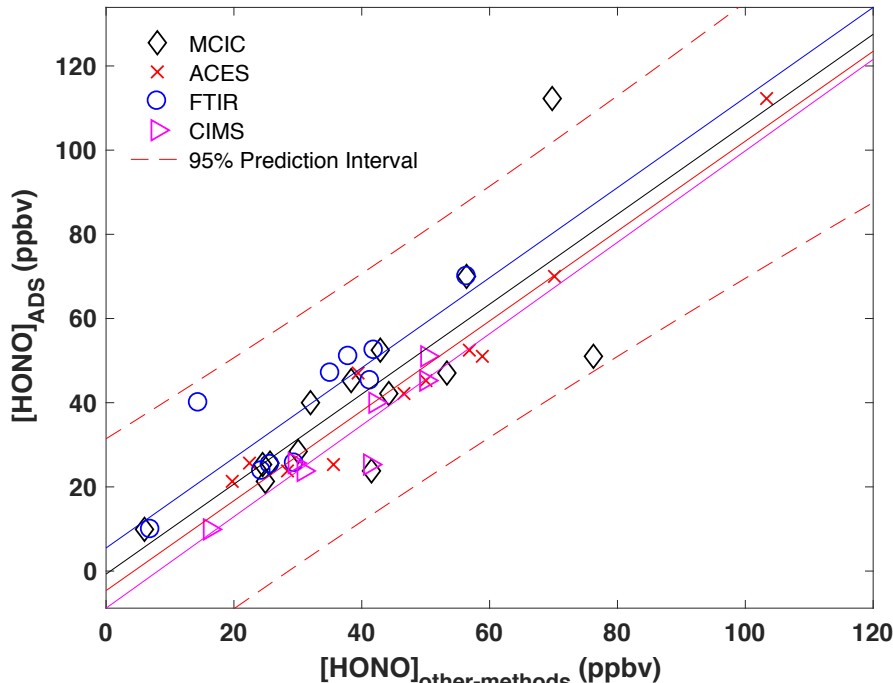

**Figure 3.** Comparison of ADS measured HONO concentration with mean values of various high resolution methods including MC/IC, FTIR, ACES and PTR-ToF for available fires. Solid lines are linear regression of each dataset with the same symbol color.

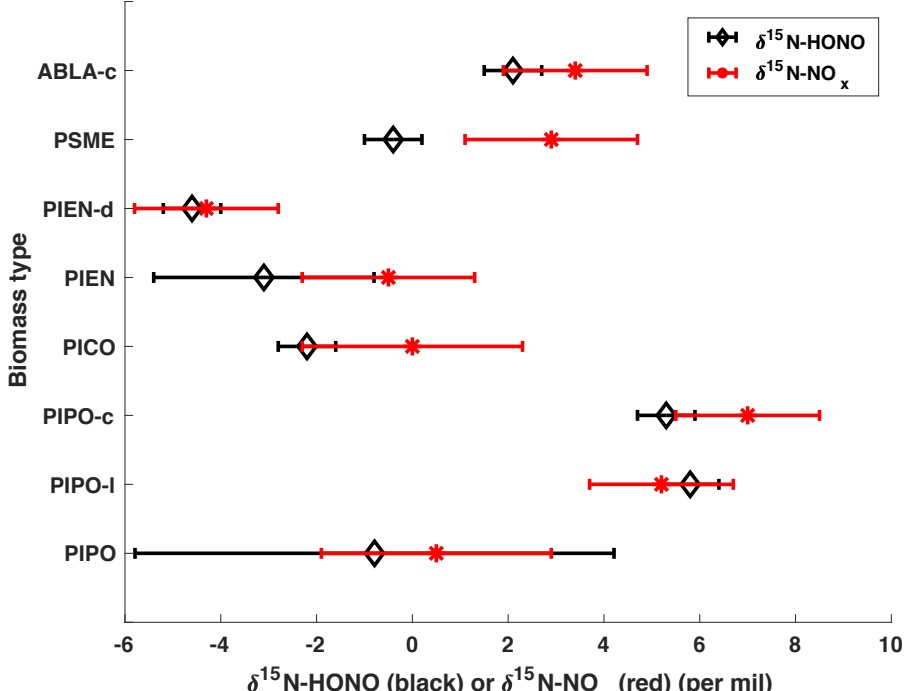

**Figure 4.** Concentration weighted mean $\delta^{15}$N- of HONO and NO$_x$ versus biomass type. The error bars are propagation of replicate $\pm 1\sigma$ uncertainty (when n>1) and method uncertainty; otherwise, the error bars stand for method uncertainty. PIPO is ponderosa pine, PICO is lodgepole, PIEN is Engelmann spruce, PSME is Douglas-fir, ABLA is subalpine (from Fish Lake, canopy). l indicates litter, c indicates canopy, d indicates duff.



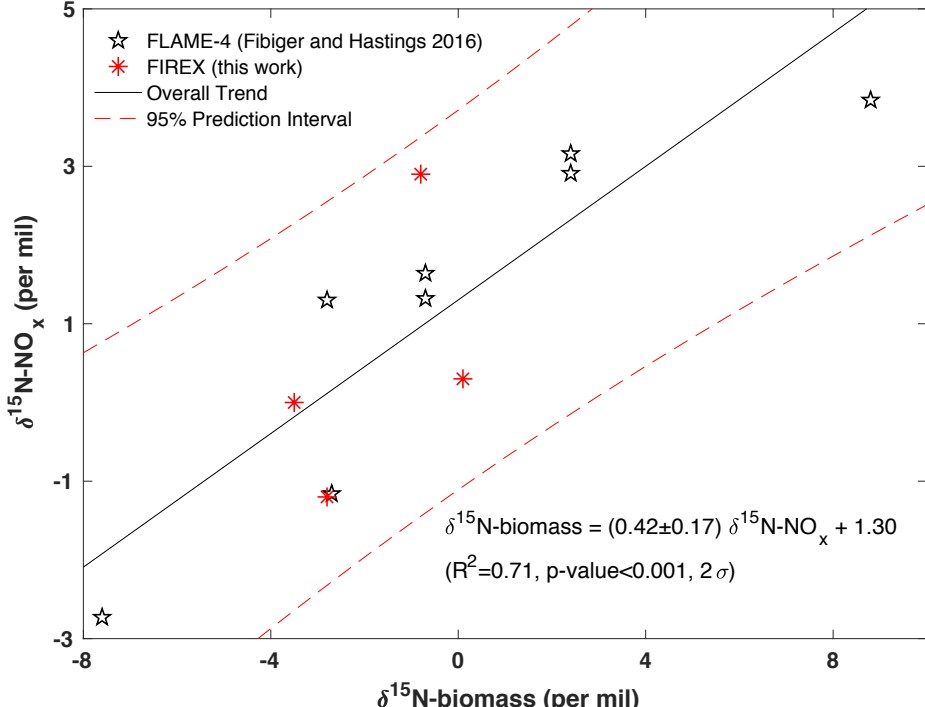

**Figure 5.** Dependence of $\delta^{15}$N-NO$_x$ on $\delta^{15}$N-biomass. Star data points represent results from FLAME-4 study (Fibiger and Hastings, 2016); Asterisk data points represent results from this work; solid line is linear regression between $\delta^{15}$N-NO$_x$ and $\delta^{15}$N-biomass for the combined dataset; dashed lines are 95% prediction interval (2σ).



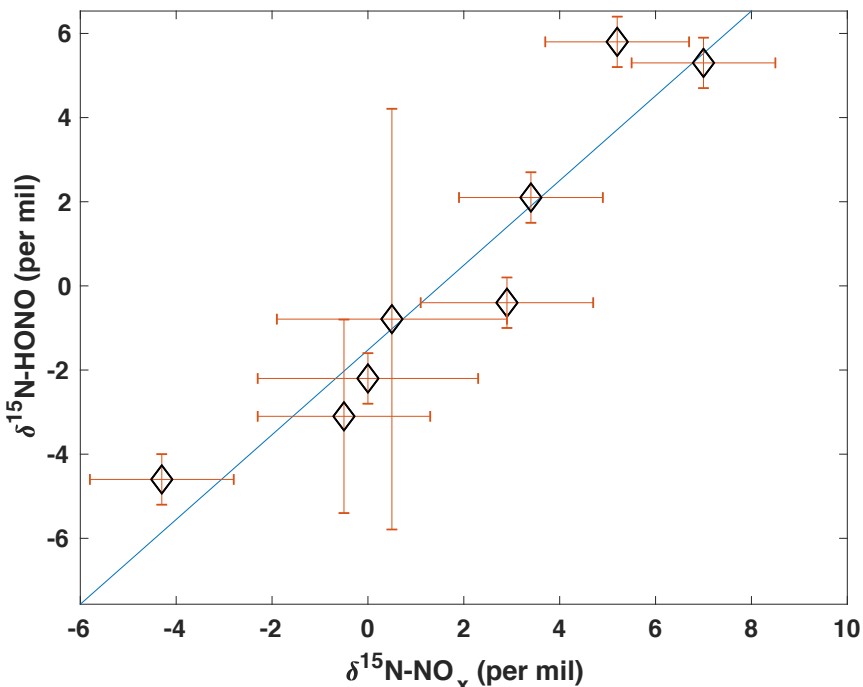

**Figure 6.** Scatter plot between $\delta^{15}$N-HONO and $\delta^{15}$N-NO$_x$. All error bars are propagation of replicate uncertainty ($\pm 1\sigma$) and method uncertainty. Linear regression follows $\delta^{15}$N-HONO = 1.01 $\delta^{15}$N-NO$_x$ - 1.52 ($R^2$ = 0.89, p<0.001).