# Peer review of "Isotopic characterization of nitrogen oxides (NOx), nitrous acid (HONO), and nitrate (NO3-(p)) from laboratory biomass burning during FIREX"

_Atmospheric Measurement Techniques, 2019_

## Referee Comment (RC1) · Anonymous Referee #1 · 6 Aug 2019

(I apologize in advance for not figuring out the LaTeX commands for "del" and "per mil" symbols for my comments.)

This paper discusses the results of measurements of the isotopic composition of NOx, HONO, and particle nitrate emission from laboratory biomass burns done as part of the FIREX campaign. This represents the first measurements of the isotopic composition of HONO from biomass burning, a critical component of the smoke radical budget in the ambient atmosphere. They find that the isotopic composition of the HONO and NOx is well corelated, suggesting they are formed via similar pathways.

The paper is very well-written and while I am not competent to evaluate the analytical

chemistry methods used, they are described thoroughly in the paper and it is clear the authors put a lot of thought into their experimental setup. The results are clearly explained and the conclusions follow from the evidence presented in the paper. Overall, this is a very good manuscript and should be published after minor revisions to address my minor concerns listed below.

L20: Missing and "and" before "proton-transfer-reaction"

L31: "connected via formation pathways" is a very vague phrase. Can you be more specific about how they are linked here?

L33: It seems odd to mention the importance of your particle nitrate measurements here, since you didn't think they were important enough to mention the quantitative results in the previous paragraph. Can you please add those results to the abstract?

L38-39: Why are the del18O measurements for each species expected to track with the influence of ozone, photochemistry and nighttime chemistry? That isn't obvious to me.

L49: While everyone knows what cooking and heating are, "prescribed" is an uncommon term and should be defined. I'd also refer to "prescribed burning" to match the "ing"s in the other terms.

L93-96: This sentence is hard to understand, with many "and"s stringing things together. Can you please rephrase, maybe splitting into two sentences to make the meaning clearer?

L152-154: I have no idea how these acronyms were made – how did "subalpine fir" become "ABLA"? Not critical as you define them, but it was a strange choice that left me constantly flipping back to understand what was burning.

L236-237: Please describe how you corrected the NOx for HONO interference, as you say on L235 that HONO is partially, not completely, converted on the catalyst.

L341-342: I'm assuming the bacteria preserves the isotope ratios and/or you can correct for the effect of the bacteria, but that isn't obvious to me, so you might want to clarify that here.

L388-394: Your explanation for the low HONO values in Fire #12 being linked to smoldering combustion and lower MCE (0.87) isn't very convincing, as Fire #15, which you note on L385-387 had the highest HONO, had an MCE of 0.89, as did Fire #17. My guess is you don't have enough information to really explain why Fire 12 was anomalously low in HONO, but in any case, you need to revise this section to include the caveat about the MCE of Fire #15.

L435 and elsewhere: Since you have both positive and negative values, I'd use the word "to" instead of the n-dash symbol to connect the ranges. i.e. "-4.3 per mil to +7.0 per mil" instead of "-4.3 per mil - +7.0 per mil."

Data availability: Consider using the CERN Zenodo archive (zenodo.org) or similar free service to store the data in a public repository with a unique DOI.

Figure 1 caption: Mention that the HONO and HNO3 in these plots was measured using the MC/IC method from the text

Table S1 caption: Please make a more descriptive caption.

Table S2 caption: Please explain why some data are missing (below detection limit? Instrument error?)

Figure S1 caption: Please clarify what the p values are for, e.g., the slope of a linear correlation?

Figure S2 caption: The caption is hard to understand. Try "Linear regression between (a) del15N-HONO and del15N-biomass (equation) and (b)..." instead.

---

## Author Comment (AC1) · 24 Oct 2019

We are grateful to Anonymous Referee #1 for the time and useful comments. We have made point-to-point responses to each question in blue text below. Please note that line numbers correspond to the revised manuscript.

For the Reviewer's and Editor's convenience, we combined our response, revised manuscript and revised supplemental information into the pdf file uploaded in the supplement area. Please let us know if there is any questions.

Thank you!

Please also note the supplement to this comment:
https://www.atmos-meas-tech-discuss.net/amt-2019-229/amt-2019-229-AC1-supplement.pdf

---

## Author Comment (AC2) · 24 Oct 2019

Thank you to Anonymous Referee #2 for the kind summary and insightful questions. We appreciate the time in reviewing our manuscript and have made point-to-point responses to each question in blue text in the attached supplement.

For the Reviewer's and Editor's convenience, we combined our response, revised manuscript and revised supplemental information into the pdf file uploaded in the supplement area. Please let us know if there is any questions.

Thank you!

Please also note the supplement to this comment:
https://www.atmos-meas-tech-discuss.net/amt-2019-229/amt-2019-229-AC2-supplement.pdf

---

## Author Response (AR1)

Authors' response: We are grateful to Anonymous Referee #1 for the time and useful comments. We have made point-to-point responses to each question in blue text below. Please note that line numbers correspond to the revised manuscript.

(I apologize in advance for not figuring out the LaTeX commands for "del" and "per mil" symbols for my comments.)

This paper discusses the results of measurements of the isotopic composition of NOx, HONO, and particle nitrate emission from laboratory biomass burns done as part of the FIREX campaign. This represents the first measurements of the isotopic composition of HONO from biomass burning, a critical component of the smoke radical budget in the ambient atmosphere. They find that the isotopic composition of the HONO and NOx is well corelated, suggesting they are formed via similar pathways.

The paper is very well-written and while I am not competent to evaluate the analytical chemistry methods used, they are described thoroughly in the paper and it is clear the authors put a lot of thought into their experimental setup. The results are clearly explained and the conclusions follow from the evidence presented in the paper. Overall, this is a very good manuscript and should be published after minor revisions to address my minor concerns listed below.

L20: Missing and "and" before "proton-transfer-reaction"

Authors' response:

Corrected. Thank you!

L31: "connected via formation pathways" is a very vague phrase. Can you be more specific about how they are linked here?

Authors' response:

Modified. Please see L23-24, "NOx and HONO are connected via formation pathways" is changed to "HONO is directly formed via subsequent chain reactions of $NO_x$ emitted from biomass combustion"

L33: It seems odd to mention the importance of your particle nitrate measurements here, since you didn't think they were important enough to mention the quantitative results in the previous paragraph. Can you please add those results to the abstract?

Authors' response:

We added the isotopic composition results for particulate nitrate in the abstract. Please see L24-26.

L38-39: Why are the del18O measurements for each species expected to track with the influence of ozone, photochemistry and nighttime chemistry? That isn't obvious to me.

Authors' response:

$O_3$ has uniquely high $\delta^{18}O$ value (~110‰) that can be distinguished from other oxidants including $O_2$, OH, and $HO_2$ (< ~20‰), therefore formation pathways of secondary products such as $NO_2$, HONO, $HNO_3$ and particulate nitrate may be distinguished with $\delta^{18}O$ value of these products. For example, $NO_2$ formed from $NO+O_3$ will exhibit much higher $\delta^{18}O\text{-}NO_2$ than that formed from $NO+HO_2/RO_2$. Nighttime $HNO_3$ is mainly formed from $NO+O_3 \rightarrow NO_2$, $NO_2+O_3 \rightarrow NO_3$, $NO_2+NO_3 \rightarrow N_2O_5$, and $N_2O_5+H_2O \rightarrow 2HNO_3$. Therefore, $HNO_3$ formed during the night has higher $\delta^{18}O$ than during the day.

L49: While everyone knows what cooking and heating are, "prescribed" is an uncommon term and should be defined. I'd also refer to "prescribed burning" to match the "ing"s in the other terms.

Authors' response:

Corrected and "prescribed burning" is defined in the text (L49). Thanks!

L93-96: This sentence is hard to understand, with many "and"s stringing things together. Can you please rephrase, maybe splitting into two sentences to make the meaning clearer?

Authors' response:

This sentence has been split in to two sentences as shown in L94-97. Thanks!

L152-154: I have no idea how these acronyms were made – how did "subalpine fir" become "ABLA"? Not critical as you define them, but it was a strange choice that left me constantly flipping back to understand what was burning.

Authors' response:

This is a good point. The acronyms are defined based on Latin names of the vegetation species, as you can see in this example (https://plants.usda.gov/core/profile?symbol=PICO)

For readers' convenience, all the acronyms have been replaced by corresponding names.

L236-237: Please describe how you corrected the NOx for HONO interference, as you say on L235 that HONO is partially, not completely, converted on the catalyst.

Authors' response:

Thank you for asking this question. "Partially" was used in the original manuscript to discuss the NOy interference species in a qualitative and conservative way. The catalytic

efficiency of NOy species other than NO2 to NO were not measured for this particular Thermo NOx analyzer, however a number of previous works found the HONO (e.g. Febo et al 1995) conversion efficiency is high and deemed the conversion efficiency 100%. We correct the NOx concentration by subtracting mean HONO concentration during each period from NOx concentration, and this provided an approximate lower limit of NOx concentration with upper limit of HONO concentration.

L341-342: I'm assuming the bacteria preserves the isotope ratios and/or you can correct for the effect of the bacteria, but that isn't obvious to me, so you might want to clarify that here.

Authors' response:

The bacteria method can preserve the isotopic signature of $NO_3^-$ and/or $NO_2^-$ so I replace "quantitative" with "complete" in the original sentence (L321). Thanks for the suggestion.

L388-394: Your explanation for the low HONO values in Fire #12 being linked to smoldering combustion and lower MCE (0.87) isn't very convincing, as Fire #15, which you note on L385-387 had the highest HONO, had an MCE of 0.89, as did Fire #17. My guess is you don't have enough information to really explain why Fire 12 was anomalously low in HONO, but in any case, you need to revise this section to include the caveat about the MCE of Fire #15.

Authors' response:

This is a really great question and thank you!

We agree with the reviewer's comment on MCE and we note for fire #12 the strong smoldering resulted in clogging of the inlet filter of the MC/IC that significantly impact the measured HONO concentration by MC/IC. Accordingly, we added the following (L369-372 and L378-382 respctively):

"We note that fire no. 12 has the smallest MCE value of 0.868 (FIREX, 2016), and an abnormal flow rate (less than half of the typical flow rate during all other measurements) due to the inlet filter clogging from extraordinarily large particulate loadings."

"Although fires no. 15 and no. 17 have relatively low MCE (~0.89), the pulse of HONO in first 5-10 minutes suggest an active flaming phase followed by longer smoldering phase. This indicates both fires had combustion conditions that consisted of a mixture of flaming and smoldering, and thus significant HONO was still produced."

L435 and elsewhere: Since you have both positive and negative values, I'd use the word "to" instead of the n-dash symbol to connect the ranges. i.e. "-4.3 per mil to +7.0 per mil" instead of "-4.3 per mil - +7.0 per mil."

Authors' response:

All "-"s have been replaced by "to" when isotopic ranges are discussed. Thanks!

Data availability: Consider using the CERN Zenodo archive (zenodo.org) or similar free service to store the data in a public repository with a unique DOI.

Authors' response:

Thank you for this information! As expected by the funding of our project, our data are archived in the NOAA database, which is also public.

Figure 1 caption: Mention that the HONO and HNO3 in these plots was measured using the MC/IC method from the text

Authors' response:

Corrected. Thanks!

Table S1 caption: Please make a more descriptive caption.

Authors' response:

A new descriptive caption has been added to Table S1. Thank you!

Table S2 caption: Please explain why some data are missing (below detection limit? Instrument error?)

Authors' response:

The missing data are results of instrumental issues and this has been clarified in the caption of Table S2. Thank you!

Figure S1 caption: Please clarify what the p values are for, e.g., the slope of a linear correlation?

Authors' response:

The p values are for the slope of a linear correlation. This has been clarified in the caption of Figure S1 (now Figure S2). Thank you!

Figure S2 caption: The caption is hard to understand. Try "Linear regression between (a) del15N-HONO and del15N-biomass (equation) and (b). . ." instead.

Authors' response:

The caption of Figure S2 (now Figure S3) has been modified based on the reviewer's suggestion. Thank you!

**Anonymous Referee #2**

Authors' response: Thank you to Anonymous Referee #2 for the kind summary and insightful questions. We appreciate the time in reviewing our manuscript and have made point-to-point responses to each question in blue text below.

Summary: The authors applied the recently developed annular denuder system (ADS) to the collection of HONO and NOx during the FIREX laboratory burns in 2016. They compare measurements of HONO across five different methods, showing decent agreement and suggesting good collection efficiency in the ADS method. Values were presented for d15N-NOx, d15N-HONO, d18O-HONO, d15N-pNO3-, and d18O-pNO3-. Some of these values compared favorably with previous measurements, and others were the first of their kind. I found this manuscript to be well written. I recommend it for publication after some minor corrections and explanations.

Specific Comments:

Abstract: I personally find that your abstract is too broad. In the first two paragraphs, there is a lot of text about describing the methods, and the real results of your work are getting buried in the later paragraphs. I think you could increase the impact of your work by removing some of the general statements and focusing on specific results (numbers and direct conclusions) as described in the latter two paragraphs.

Thank you, the abstract has been modified following the reviewer's suggestions.

Pg 5 Ln 152-153: How did you decide on the acronyms/abbreviations for each fuel type? Most of them don't seem to make any intuitive sense, and this is very distract- ing when trying to understand your results. I'd recommend you make them easier to understand, e.g., Douglas-fir (DFIR), etc.

Authors' response:  Thank you for the suggestion! The acronyms are defined based on Latin names of the vegetation species, as you can see in this example (https://plants.usda.gov/core/profile?symbol=PICO)

For readers' convenience, all the acronyms have been replaced by corresponding names.

Pg 6 Ln 225: I have several questions regarding this section about the NOx measure-ments: a) Throughout the manuscript, whenever you give NOx values or use NOx values in calculations, are you using the values from the collection of NOx after the ADS system, or are you using NOx values measured using the Thermo instrument? My impression is that you always use the former, and never show any data from the Thermo instrument. If so, you really have no reason to mention the Thermo measurement at all here, and it could be removed. (If not, then please be explicit throughout the manuscript about where the NOx data originated at each mention in the text). b) However, I don't

suggest you remove the description of the Thermo instrument! Instead, I suggest you show the data that underlies the statement "The NOx measurement verified the concentration of the NOx collected for isotopic analysis, and the original NOx data is available. . ." This would be a plot analogous to Fig. 3 but for NOx, potentially in a new section analogous to Sect 3.2 (or in the SI, in which case the Thermo instrument description should also move there).

Authors' response:

Thanks for bring up this question and the suggestion! The NOx concentrations we used in discussion are all from the NOx collection system, as it offers a direct comparison with HONO concentration in the same system. The NOx analyzer measurement is used to verify the NOx collected with our collection system and the original 1-minute time resolution data is stored in NOAA archive. Since the NOx collection method has been validated in both lab and field previously (Fibiger et al., 2014, Fibiger and Hastings, 2016), we did not expand it in this manuscript. Apologies for the confusion. Based on the reviewer's suggestion, we add Table S1 and Figure 1S to show the comparison of [NOx_collected] versus [NOx_analyzer], in addition to moving the NOx analyzer description to supplemental information. Please note that each NOx concentration recovered from the collection system represents mean concentration across each whole fire, and therefore the [NOx_analyzer] concentrations used for the comparison are a mean value across the whole fire.

Pg 6 Ln 234-237: How important is the interference of HONO on the Thermo NOx instrument? Please provide more information about how you corrected for it (e.g., what fraction of HONO did you assume gets converted to NO in the catalyst?), and provide the numbers for how large the correction was relative to the total NOx measurement. At this point, you only mention the correction in passing, and a reader cannot determine how important the correction is.

Authors' response:

Thank you for asking this question. HONO is an important interference with NO2 using the chemiluminescence analyzer. We correct the NOx concentration by subtracting mean HONO concentration during each fire from NOx concentration by assuming HONO is 100% converted to NO on Molybdenum catalyst (e.g. Febo et al 1995), and this provided an approximate lower limit of NOx concentration. The following text is added to the description section of the NOx analyzer, which is now moved to SI.

"In this study, only the HONO interference was corrected for. This was done by subtracting the ADS measured HONO concentration (mean value across each whole fire) from Thermo analyzer measured $NO_x$ concentration averaged across the whole fire; this provided the approximate lower limit of the $NO_x$ concentration by assuming HONO is 100% converted to NO on the Molybdenum catalyst (e.g. Febo et al 1995)."

Pg 9 Ln 330: It should be mentioned somewhere here that the PTR-ToF is a mass spectrometer. It is important for consistency that you also use the same acronym for the

instrument as they use in the cited Koss and Yuan papers.

Authors' response:

Thanks for the suggestions. The PTR-ToF mass spectrometer is briefly described in the next sentence following Ln 330.

Pg 10 Ln 394: Your HONO/NOx ratios were calculated as averages across the entire burn, correct? This is good for comparison to previous results, but I'm wondering if you can also calculate a time dependent HONO/NOx ratio across each burn. In other words, as each burn transitions from flaming to smoldering, does this HONO/NOx ratio change? This would be important to note for real wildfires that may have different ratios of flaming vs smoldering emissions.

Authors' response:

This is a very interesting comment! HONO/NOx ratios reported here are based on the concentrations collected in our HONO and NOx collection train. The time resolution of our data is over the whole fire period; similarly, MCE values (from CO and CO2) are also reported by fire (in the NOAA FIREX Firelab data archive). The primary focus of this manuscript is isotopic composition analyses of reactive nitrogen species from biomass burning, and the higher time resolution HONO/NOx ratio vs MCE could be an interesting topic for future work. Lastly, since both HONO and NOx are favored by flaming combustion condition, we would not expect to see a strong trend of the relation between HONO/NOx vs MCE.

Pg 14 Ln 567: This line suggests that the average combustion temperature was "moderate" (e.g. 700C) rather than high temp (>850C). How does that translate to flaming vs smoldering? What is a typical temperature for each phase? More specifically, my question is are you saying that the flaming during FIREX was typically closer to 700C than 850C, or are you saying that flaming could have been near the hotter end of the spectrum but the dominant source of emissions was from smoldering processes at lower temp?

Authors' response:

Great questions. Our results do not suggest the average combustion temperature was "moderate" (e.g. 700C) rather than high temp (>850C); instead, our results only suggest R1-R6 occurred to a larger extent than R7-R11 (percentage can be used as an analogy). If R1-R6 is dominant over R7-R11, or say R7-R11 is negligible, R1-R6 would result in much larger isotopic effect, leading to much larger $\delta^{15}N\text{-}NO_x$ than $\delta^{15}N\text{-}HONO$ (e.g. by >10‰) instead of the small difference observed here (Figure 4 and Figure 6). … Biomass combustion temperature ranges from 200°C—1200°C with flaming at 1000-1200°C, glowing at ~800°C and smoldering <600°C, and the temperature changes as fire progresses. Many fires are mixtures of flaming and smoldering, and therefore we do not intend to infer flaming or smoldering with our comparison of $\delta^{15}N\text{-}NO_x$ with $\delta^{15}N\text{-}$ HONO. Rather, this comparison suggests a source signature that could be used to track

wildfire emission and its aging process in the atmosphere.

General comment: Related to my previous comment, can you add a few sentences (perhaps in the conclusions section) about how exactly your measurements of fresh lab smoke may relate to measurements of fresh wildfire smoke? You've given some discussion about how you expect the d15N-HONO and d15N-NOx to depend on combustion temperature, and in wildfires you are likely to have instances where the (average) combustion temperature is potentially very different than what you measured in the lab. To my understanding, there is currently no good way of determining exact combustion temperatures either in the lab or in the field, so I'm not asking you to solve this problem but rather to acknowledge it and provide some guidance for how future isotopic measurements in wildfires could be interpreted relative to your results.

Authors' response:

We agree with the reviewer regarding this point. In addition to our response to the previous question, we have modified original text in the conclusion section to now read as follows (Ln 640-649):

"The relationship between $\delta^{15}$N-HONO and $\delta^{15}$N-NO$_x$ likely reflects that HONO was produced to a larger extent at moderate combustion temperatures (< ~800 °C) than higher temperatures on the basis of a simplified mechanism for flow of reactive nitrogen species. However, we note that this relationship is derived from all measured $\delta^{15}$N-HONO and $\delta^{15}$N-NO$_x$ in fires ranging from smoldering to flaming, so is not necessarily representative of a particular combustion condition. Still, it is likely that a compilation over a range of conditions is more useful for potentially distinguishing HONO sources and formation pathways in the environment since it will always be a challenge to assess exact combustion temperatures."

Figure 3: Please change the "CIMS" label in the legend to match the "PTR-ToF" designation in the caption.

"PTR-ToF" has been put in the legend to replace "CIMS".